# Identification and Characterization of Elevated Expression of Transferrin and Its Receptor TfR1 in Mouse Models of Depression

**DOI:** 10.3390/brainsci12101267

**Published:** 2022-09-20

**Authors:** Xin Chang, Mengxin Ma, Liping Chen, Zhihong Song, Zhe Zhao, Wei Shen, Huihui Jiang, Yan Wu, Ming Fan, Haitao Wu

**Affiliations:** 1School of Basic Medical Sciences, Anhui Medical University, Hefei 230032, China; 2Department of Neurobiology, Beijing Institute of Basic Medical Sciences, Beijing 100850, China; 3School of Information Science & Engineering, Lanzhou University, Lanzhou 730000, China; 4Key Laboratory of Neuroregeneration, Co-innovation Center of Neuroregeneration, Nantong University, Nantong 226019, China

**Keywords:** depression, mental disorders, proteomics, chronic social defeat stress (CSDS), transferrin (TF), transferrin receptor 1 (TfR1), blood-brain barrier (BBB), bioinformatics

## Abstract

Depression has become one of the severe mental disorders threatening global human health. In this study, we first used the proteomics approach to obtain the differentially expressed proteins in the liver between naive control and chronic social defeat stress (CSDS) induced depressed mice. We have identified the upregulation of iron binding protein transferrin (TF) in the liver, the peripheral blood, and the brain in CSDS-exposed mice. Furthermore, bioinformatics analysis of the Gene Expression Omnibus (GEO) database from various mouse models of depression revealed the significantly upregulated transcripts of *TF* and its receptor *TfR1* in multiple brain regions in depressed mice. We also used the recombinant TF administration via the tail vein to detect its permeability through the blood-brain barrier (BBB). We demonstrated the permeability of peripheral TF into the brain through the BBB. Together, these results identified the elevated expression of TF and its receptor TfR1 in both peripheral liver and the central brain in CSDS-induced depressed mice, and peripheral administration of TF can be transported into the brain through the BBB. Therefore, our data provide a compelling information for understanding the potential role and mechanisms of the cross-talk between the liver and the brain in stress-induced depression.

## 1. Introduction

Depression is one of the most prevalent mental disorders caused by genetic, environmental, psychological, and biochemical factors [1,2,3]. It is projected to be the leading cause of disease burden worldwide by 2030 [4]. Patients with depression usually suffer from long-term sadness and anhedonia. In severe cases, suicidal thoughts and behaviors may occur. At present, the pathophysiological mechanisms of depression are still challenging due to the complicated symptomatology, which lead to the limited efficacy of existing treatments [5,6]. In the past few decades, the pathogenesis of depression has been extensively studied through gene sequencing technology, proteomics technology, and biopsychosocial perspective [7,8,9]. There is increasing evidence that molecular mechanisms and cascades are involved in the pathology of depression such as low-level neurotrophic factors, in particular, brain-derived neurotrophic factor (BDNF) [10], chronic inflammation [11], the hypothalamic-pituitary-adrenal (HPA) axis deregulation [12], etc. [13]. Moreover, the structural and functional abnormalities of specific brain regions, such as amygdala, hippocampus, dorsomedial thalamus, and prefrontal cortex, and impaired brain circuits have been extensively identified in either major depressive disorder (MDD) patients or depressive animal models [14,15,16,17].

So far, the majority of studies on the mechanisms of depression mainly focus on the brain, rarely on the peripheral tissues. Recent studies proposed a useful framework reflecting the anatomical-functional interplay between the brain and peripheral organs in fear conditioning and high-level cognitive impairments [18,19]; this opens the door towards identifying a possible psychophysiological biomarker of the individual proneness towards depression [18]. Interestingly, traditional Chinese medicine states that “too much fire in the liver makes you feel aggressive and too reactive”, which also indicates the close interrelationship between liver metabolism and stress-induced negative emotions [20,21,22]. Although the liver is a very important organ, how its dysfunction affects the emotional and mental activities remains largely elusive [23,24].

The chronic social defeat stress (CSDS) model is one of the commonly used animal models to study depression. Compared with other animal models of depression, CSDS model has social stressors, which can better imitate the pathogenic process of human social depression [25]. Moreover, the modeling and evaluation methods of the preclinical CSDS rodent model have been further optimized in the past few years, and make it a reliable animal model of depression [26,27]. CSDS stressed mice tend to spend less time struggling against inescapable stressors such as forced swimming or tail suspension. Furthermore, the stress-susceptible mice also display a robust depression-like phenotype marked by reduced social interaction, increased anhedonia, and significant body-weight changes [28]. Based on these criteria, we are able to screen and identify the potential novel signaling molecules involved in the CSDS-induced depression in both the liver and the brain.

In this study, the CSDS mouse model was successfully generated, and the proteomic analysis of the livers from the stress induced susceptible and control mice was carried out through two-dimensional electrophoresis (2-DE) combined with mass spectrometry analysis. The differentially expressed proteins in susceptible mice were identified and verified by quantitative real-time PCR, Western blot, and the enzyme linked immunosorbent assay (ELISA), respectively. Interestingly, we found that the expression of transferrin (TF) was significantly up-regulated not only in the liver, but also in the peripheral blood, and multiple brain regions. Moreover, based on the GEO database of RNA-sequencing data from various stress rodent models, in agreement with our results, the transcriptional levels of transferrin and its receptor TfR1 were both dramatically increased in brain and peripheral organs. Importantly, we also demonstrated that recombinant TF peripherally injected via the tail vein can be transported into various organs including the central nervous system, probably through the blood brain barrier (BBB). Together, this study provides a novel insight into the potential roles of TF in the liver and its receptor TfR1 in the brain in stress-induced depression.

## 2. Materials and Methods

### 2.1. Chronic Social Defeat Stress Mice Model

Male C57BL/6 mice (6–8 weeks old, *n* = 50) and male CD-1 mice (7 months old, *n* = 20) were purchased from Beijing SPF Biotechnology company. C57BL/6 mice are the most commonly used inbred mouse strain for in vivo study because of their genetic homogeneity, reproducibility of results, inexpensive cost and availability of a wide array of research reagents [29]. The construction method of CSDS model is described in a previous literature report [30].

Briefly, the CD-1 mice in a single cage were raised for 7 days to produce territorial awareness, and the C57BL/6 mice were raised for 7 days to adapt to the environment. First, the aggressiveness of the CD-1 mice waw screened. The C57BL/6 mice were put in the cage of the CD-1 mice and counted for 3 min. If the CD-1 mice attack more than 5 times and the time for the first attack is less than 1 min, the CD-1 mice are considered to be aggressive; the mice are considered aggressive if this occurs at least 2 times in the 3-day aggressive screening. Therefore, they can be included in the experiment as the attacking mice of the model. Then. the mice in the experimental group underwent CSDS for 10 days. That is, the C57BL/6 mice in the experimental group were placed in the CD-1 mouse cage every day for 10 consecutive days. After being challenged for 5–10 min, they were isolated with a transparent acrylic plate with small holes for 24 h. The CD-1 mice were replaced every day, so as to avoid the decrease of aggression caused by familiarity between C57BL/6J mice and CD-1 mice. After 10 days, the C57BL/6 mice in the experimental group (*n* = 19) were taken out and kept in a single cage. The non-stressed control is C57BL/6 mice of the same age (*n* = 19). After being adapted to the same environment, the control mice were exposed to non-aggressive C57BL/6 mice for 5–10 min every day, and then the same perforated transparent acrylic plate was used as previously described. The control mice were isolated for 24 h and replaced with new C57BL/6 mice every day. After 10 days, the non-stressed control C57BL/6 mice were taken out and kept in a single cage.

All animals were group housed in standard conditions on a 12-h light/dark cycle with access to food and water. All animal experiments were conducted in accordance with protocols approved by the Institutional Animal Care and Use Committee of the Beijing Institute of Basic Medical Sciences.

### 2.2. Behavioral Experiments

#### 2.2.1. Social Interaction Test

The social interaction test was carried out according to the previous literature report [30]. The mice were placed in an open field (41 cm×41 cm), and there was a metal mesh cage (10 cm × 8 cm) on one side of the open field. The test consisted of two stages, each for 2.5 min. In the first stage, there was no target in the metal cage net, and the CSDS-induced stressed mice and non-stressed control C57JBL/6 mice were placed in the open field and allowed to move freely (*n* = 19 for each group). In the second stage, a new CD-1 mouse was placed in a metal cage, and the C57JBL/6 mouse was returned to the open field. During the experiment, the ANY-maze (Stoelting, Chicago, USA) software was used to record the time the mice spent in the social interaction area (diameter 16 cm) and the corner area (8 cm × 8 cm). The social interaction ratio (SI) was calculated as time spent in the interaction zone in the second stage trial divided by time spent in the interaction zone in the first stage trial. It is believed that the mice with SI < 1 show social avoidance as being of the susceptible type, and the mice with SI > 1 show social preference as a resistant type.

#### 2.2.2. Sucrose Preference Test

The test was slightly modified based on a previous literature report [31]. First, the non-stressed control mice and CSDS-induced susceptible mice (SI < 1) (*n* = 12 for each group) were given two bottles of water for 24 h to adapt to the two bottles for drinking, then one of the bottles of water was changed to a 1% sucrose solution, and the weight of the two bottles was weighed and recorded. After 12 h, the two bottles were replaced. And, the two bottles were weighed again after 24 h of testing. The sucrose preference is calculated as a percentage of the weight of sucrose water consumed over the total weight of fluid consumption.

#### 2.2.3. Forced Swimming Test

The forced swimming test is a classic experiment used to evaluate CSDS-induced depression-like behaviors in animals [32]. The experimental device was a 3 L beaker, into which 2 L of distilled water was added, and the water temperature was 22 ± 1 °C. The distilled water should be changed between the two animals. On the first day of the experiment, the non-stressed controls and CSDS-induced susceptible mice (*n* = 12 for each group) were put into a beaker, and after swimming for 15 min, they were taken out and dried. On the second day of the experiment, those mice were placed in a beaker and timed for 6 min, and the immobility time of the next 4 min was recorded.

#### 2.2.4. Tail Suspension Test

The tail suspension test is also a classic experiment for evaluating CSDS-induced depression-like behaviors in animals [33]. A rubber band was used to fix the mouse’s tail 3 cm and hung 30–40 cm above the ground for 6 min, and the immobility time of the next 4 min was recorded in CSDS-induced susceptible mice and non-stressed controls (*n* = 12 for each group).

### 2.3. Two-Dimensional Electrophoresis and Mass Spectrometry

A total of 10 mg of liver tissue was taken, 30 uL protein lysate (Cat# 89900; Sigma-Aldrich, St. Louis, MO, USA) was added, and fully ground and lysed on ice. The protein supernatant was sucked out slowly, the protein concentration was determined by BAC method (Cat# 23227; Sigma-Aldrich, St. Louis, MO, USA), and the sample after quantification was loaded. The two-dimensional electrophoresis (2-DE) gel proteins were reduced in order to break disulphide bonds and alkylated to prevent the bonds reforming as previously described [34]. 2-DE was slightly improved according to the method reported in the literature [35]. Protein 800 μg samples were taken and were actively hydrated at 50 V for 16 h. The boost procedure is 250 V linear boost for 30 min, 1000 V fast boost for 1 h, 10,000 V linear boost for 5 h, and 10,000 V fast boost to 70,000 vh. After isoelectric focusing, the adhesive strip was balanced. Then, the separation was carried out at a constant current of 60 mA, and the gel concentration was 12% of the sodium dodecyl sulfate polyacrylamide gel electrophoresis (SDS-PAGE) (2-DE reagents are purchased from Bio-Red, Hercules, CA, USA). 2-DE pattern was obtained after use of silver nitrate staining and gel imaging system. The differential protein spots were cut from the 2-DE gel electrophoresis and placed in a 0.5 mL EP tube; the sample was treated according to the method in document [36]. MS/MS mass spectrometry analysis was performed on the micromass ESI-Q-TOF Ultima global mass spectrometer (Waters, Milford, MA, USA).

### 2.4. Quantitative Reverse Transcription PCR

The appropriate amount of mouse tissue should not exceed 100 mg in a 1.5 mL EP tube. In total, 500 μL TRIzol (Cat# 15596018; Invitrogen, Carlsbad, CA, USA) was added to each tube and ground thoroughly on ice carefully to prevent splashing. After the grinding was complete, 500 μL Trizol was added to each tube, shaken and mixed, and the manufacturer’s instructions were followed. According to the amount of precipitation, 20–40 μL RNAase-free distilled water was added and placed in a water bath at 65 °C to fully dissolve the precipitate. To reduce or eliminate genomic DNA contamination in RNA preparations, DNase I (Cat# 18047019; Thermo, Waltham, MA, USA) treatment was used for removing DNA contamination from RNA samples. A spectrophotometer was used to measure the RNA concentration, and the RNA quality was evaluated based on the 260/280 value. RNAase-free distilled water was added to adjust the RNA concentration of each tube to about 500 ng/μL. According to the reverse transcription kit instructions (Cat# RR036; TaKaRa, Tokyo, Japan), RNA was reverse transcribed into cDNA. The experiment was carried out based on the instructions of real-time fluorescent quantitative PCR (Cat# CW2601; CWBIO, Beijing, China). After the reaction, the relative mRNA expression level was calculated according to the CT value of each well. The cycle threshold (CT) value for the mRNA of target genes was first normalized by deducting the CT value for *Actb1* to obtain a ΔCT value. ΔCT values of test samples were further normalized to the average of the ΔCT values for control samples to obtain ΔΔCT values. Relative gene expression levels were then calculated as 2^-^^ΔΔCT^ as previously described [37,38]. Primers used for quantitative RT-PCR analysis are listed in Table 1.

### 2.5. Protein Extraction and Western Blots

An appropriate amount of mouse tissue was weighed in a 1.5 mL EP tube. A total of 30 μL of protein lysis solution (Cat# 89900; Sigma-Aldrich, St. Louis, MO, USA) was added for every 10 mg of liver tissue, and 20 μL of protein lysis solution was added for every 10 mg of brain tissue (*n* = 4 for CSDS-induced susceptible mice and non-stressed controls, respectively). After being fully ground on ice, it was lysed and centrifuged on ice. The protein supernatant was aspirated slowly, and the volume of the supernatant was recorded. The BAC method was adopted to determine the protein concentration (Cat# 23227; Sigma-Aldrich, St. Louis, MO, USA), the corresponding protein concentration was calculated according to the standard curve, and the protein concentration was adjusted to be consistent. The 5 × loading buffer in proportion was added, shaken and mixed well, and boiled in 100 °C water for 10 min. To identify protein levels, samples (20–50 μg) were electrophoresed on SDS/PAGE and transferred to PVDF membrane. 5% skimmed milk powder liquid was sealed for 1–2 h at room temperature. The primary antibody was incubated overnight at 4 °C. The secondary antibody was incubated for 1 h at room temperature. Finally, the film was developed and scanned and analyzed (Bethesda, Rockville, MD, USA). The following antibodies were used: rabbit anti-TfR1 antibody (1:500; Cat# 226290; Biocompare, South San Francisco, CA, USA), anti-transferrin antibody (1:500; Cat# ab82411; Abcam, Cambridge, MA, USA), anti-β-actin antibody (1:1000; Cat# 3700; CST, Danvers, MA, USA).

### 2.6. Enzyme Linked Immunosorbent Assay

Serum extraction: After weighing the mice, an appropriate amount of 1% sodium pentobarbital solution was injected according to their body weight. After the mice were fully anesthetized, the abdomen of the mice was placed upwards, and the limbs were fixed on the foam board. The mouse’s chest cavity was carefully cut to fully expose the heart. A 1 mL syringe was gently inserted into the right ventricle of the mouse to slowly draw blood from the mouse. The needle was removed and the blood was slowly added to a 1.5 mL EP tube at 4 °C overnight. After centrifugation, the supernatant was transferred to the EP tube slowly.

Tissue extraction: An appropriate amount of mouse tissue (*n* = 10 for CSDS-induced susceptible mice, resilient mice and non-stressed controls, respectively) was taken in a 1.5 mL EP tube and weighed. 100 μL of pre-cooled PBS solution containing protease inhibitors was added for every 10 mg and homogenized thoroughly on ice with a glass homogenizer, and the homogenate was added to the EP tube. It was centrifuged, the supernatant was aspirated, an aliquot taken, and stored at −80 °C.

According to the kit instructions (Cat# ab157724; Abcam, Cambridge, MA, USA. Cat# ml001959; Mlbio, Shanghai, China) configure 1× diluent, 1× washing buffer, and 1× enzyme-antibody conjugate was used. Other reagents were removed from 4 °C and equilibrated at room temperature. Then we prepared the standard, diluted the sample, added blank diluent, standard diluent and sample diluent to 96-well plates. After incubation at room temperature, we washed the sample, added enzyme-antibody conjugate and washed after incubation at room temperature. Finally, the substrate color developing solution and stop solution were added to terminate the reaction. We measured the absorbance at 450 nm with a microplate reader, calculated the substrate content of the corresponding sample according to the standard curve, and performed statistical analysis.

### 2.7. Red transferrin Conjugate Injection

The protein coupling solution was simply centrifuged with a microcentrifuge to eliminate any protein aggregates that may form during storage before use (Cat# P35376; Thermo, Waltham, MA, USA). The service concentration of the liquid is 25 μg/mL and the dosage is 0.2 mL. The mice (*n* = 3) were fixed with a special tail vein injection rack fixing device. Before injection, the tail was wiped repeatedly with cotton dipped in slightly hot water for a few minutes. Vasodilation is conducive to injection. Then we wiped the tail with an alcohol cotton ball, selected an appropriate needle for injection, entered the needle on an inclined plane, and advanced 1/3 of the needle length in parallel. After testing whether the needle was in the vein by looking for blood return, we finally pushed in the required amount of liquid reagent. At 3 h after injection, the mice were anesthetized, decapitated and killed, and the tissues were taken in turn. Sections of samples, with 35 μm thickness, were taken, and the red fluorescence observed under a confocal microscope.

### 2.8. Acquisition and Analysis of RNA Sequencing Data

Gene Expression Omnibus (GEO) database is a Gene Expression database created and maintained by the National Center for Biotechnology Information (NCBI). We downloaded the original RNA sequencing raw data from the GEO database by using two keywords “stress model” and “animal”. TF and TFR were retrieved from the original data, and relevant data were extracted for statistical analysis and the organization source and model name were recorded. We eventually found five key databases termed with GSE151807, GSE100236, GSE172451, GSE180055, and GSE86077, respectively, for further bioinformatic analysis.

### 2.9. Statistical Analysis

Graph Pad Prism 7 (Version X; La Jolla, CA, USA) was used for statistical analysis. The *t* test was used for comparison between the two groups. One-way analysis of variance was used to compare the three groups. Two-way analysis of variance was used for repeated measures data. *p* < 0.05 was considered to be statistically significant.

## 3. Results

### 3.1. Establishment and Evaluation of the CSDS Mouse Model

Given that CSDS mouse model can better imitate the stress state of human society than other animal models, in this study, the CSDS mouse model was generated and used to identify the potential signaling molecules in the liver involved in the stress-induced depression [25,39]. Then, 24 h after the last round of social defeat stress, both the stressed mice and the non-stressed controls were exposed to unfamiliar CD-1 mice for social interaction assay. A subset of stressed mice exhibited avoidance, with a social interaction ratio less than 1 in the social interaction assay and were designated as “susceptible” mice; the subset of stressed mice with a social interaction ratio great than 1 were designated as “resistant” mice. The results of this study showed that the social interaction ratio of non-stress control mice (*n* = 19) and susceptible mice (*n* = 19) was significantly different (P < 0.0001) (Figure 1A–C). In the social interaction experiment, the susceptible mice were significantly different in behavior from that of the control mice, and were more inclined to curl up in the corner of the open field by showing the typical phenomenon of social avoidance. Interestingly, we also found that the concentration of the serum cortisol in the susceptible mice (*n* = 10) was much higher than that in the non-stressed controls (*n* = 12, *p* = 0.0016) (Figure 1D), suggesting the significantly upregulated expression of stress-related hormones in CSDS-induced susceptible mice.

Meanwhile, the classic behavioral tests were carried out to evaluate depression-like behaviors in CSDS stressed mice, such as sucrose preference test, forced swimming test and tail suspension test. In the sucrose preference test, we found that the preference of the susceptible mice (*n* = 12) to the 1% sucrose solution was significantly lower than that of the non-stress control mice (*n* = 12) (*p* = 0.0004) (Figure 1E), reflecting that the stressed mice have an anhedonia-like phenotype. Accordingly, in the forced swimming and tail suspension tests, the immobility time of the susceptible mice increased significantly in the last 4 min compared to controls (*p* = 0.0059, *p* = 0.005) (Figure 1F,G), indicating the stressed mice have the phenotypes of desperation. Together, our data show that we have successfully generated a reliable preclinical mouse model of CSDS, which exhibits long-term physiological and behavioral phenotypes similar to depression and anxiety.

### 3.2. Identification of Differentially Expressed Proteins in the Stressed CSDS Liver by Two-Dimensional Electrophoresis and Mass Spectrometry

To identify the potential signaling molecules related to stress-induced depression in the liver in susceptible mice, the liver tissues were dissected and homogenized from both stressed susceptible mice and non-stressed controls. Next, we carried out two-dimensional electrophoresis (2-DE) and liquid chromatography mass spectrometry (LC-MS) assay to identify the differentially expressed proteins between stressed CSDS livers and non-stressed controls (Figure 2A). As shown in Figure 2B, the differentially expressed proteins between control and susceptible livers were distinguished and obtained by 2-DE. Those differential proteins were further cut, digested, and analyzed by mass spectrometry (Figure 2A).

A total of 15 protein sites were identified with significant changes, of which PGM1, PMM2 and INMT showed a downward trend, and the rest showed an upward trend (Figure 2B). Subsequently, the 15 protein sites were further analyzed by mass spectrometry. Finally, a total of 10 proteins with significant expression changes were identified in susceptible livers based on the ProtScore. As shown in Table 2, these 10 differentially expressed proteins include heat shock proteins Hspa9 and Hspa5, transferrin (TF), Carbamoyl-phosphate synthase 1 (Cps1), Phosphoglucomutase 1 (PGM1), Phosphomannomutase 2 (PMM2), Purine nucleoside phosphorylase (Pnp), Indolethylamine N-methyltransferase (INMT), High mobility group protein b1 (Hmgb1), and Amyloid P component, serum (Apcs). Among them, except for PGM1, PMM2, and INMT, the expression of the other 7 proteins were significantly upregulated in susceptible mice (Figure 2B), suggesting the potential role of these molecules in the pathogenesis of CSDS-induced depression in mice. Moreover, to further confirm the reliability of the results from 2-DE, we detected the transcripts of those differentially expressed proteins by qRT-PCR. As shown in Figure 2C, the upregulation of *transferrin* (*TF*), *Cps1*, and *Hmgb1* was also confirmed at the transcriptional level. Interestingly, it should be noted that TF is a serum-abundant metal-binding protein, which is synthesized primarily in the liver and significantly upregulated in stressed CSDS liver. However, its roles in both the liver and stress-induced depression are largely unknown. Therefore, we next mainly focus on the potential role of TF in CSDS-induced mice with depression.

### 3.3. Characterization of the Elevated Expression of Transferrin in the Liver and Blood Serum Induced by CSDS

TF, a blood-plasma glycoprotein, is one of the key molecules for absorption and transportation of iron, which plays a central role in iron metabolism. As mentioned above, TF is mainly synthesized by the liver, and the remaining small part is synthesized by ependymal, oligodendrocytes, Sertoli cells, and so on [40]. Given that our proteomics analysis indicated the upregulation of TF in the liver after CSDS, next we further verified the expression level of TF in the liver and peripheral blood serum by Western blot and ELISA techniques, respectively. Compared to the non-stressed controls, the expression of TF in susceptible mice was significantly upregulated in the liver lysates (*p* = 0.0040, *n* = 4) (Figure 3A,B). Interestingly, in addition to TF, its receptor TfR1 was also found significantly upregulated in the stressed CSDS liver in susceptible mice compared to controls (*p* = 0.0286, *n* = 4) (Figure 3A,C).

Meanwhile, to further explore the correlation between the expression level of TF and the sensitivity of CSDS-induced depression, we compared the concentration of TF in the liver and blood serum among the non-stressed control, the resilient mice, and the susceptible mice by ELISA analysis (Figure 3D). It should be noted that, in the CSDS model, the stressed mice with social index (SI) > 1 in social interaction experiment were defined as resilient mice, which showed less sensitivity to the same CSDS treatment compared to susceptible mice. Our ELISA analysis demonstrated that the level of circulated TF was also significantly upregulated in both the liver (F = 6.934, *p* = 0.0037, *n* = 10) and the blood serum (F = 14.9, *p* < 0.0001, *n* = 10) of the susceptible and resilient mice compared to control mice (Figure 3E,F). Interestingly, we also noticed that the upregulated level of serum TF in the susceptible mice was even higher than that of the resilient mice (Figure 3F), suggesting the potential correlation between the serum concentration of the TF and the associated stress-induced depression.

### 3.4. Elevated Expression of TF and Its Receptor TfR1 in Brain Regions after CSDS

TF transports iron through the blood to various tissues throughout the body, such as liver, spleen, and bone marrow. Importantly, previous studies have shown that iron-containing TF can be transported through the blood brain barrier (BBB) into the brain [41,42]. To further confirm this notion, we injected red transferrin co-junction reagent into the tail vein of the mouse, and 3 h later, the liver, heart, spleen, lung, and kidney from the injected mice (*n* = 3) were isolated and fixed for the cryosection and confocal imaging of red fluorescent signals in different tissues (Figure 4A). In agreement with previous studies, we noticed that red fluorescence conjugated TF could be detected in multiple organs through blood circulation 3 h after tail vein injection (Figure 4B,C). It should be noted that we also found the distribution of red fluorescent signals throughout the brain slides including the medial prefrontal cortex (mPFC), ventral and dorsal hippocampus, lateral habenula, and amygdala (Figure 4C). Our data further confirmed that TF could be transported from the peripheral organ, such as the liver or blood, into the brain through the blood-brain barrier (BBB) under physiological conditions.

Next, we asked whether the expression of TF is upregulated in the brain tissue of CSDS-induced susceptible mice. Then we detected the expression of TF and its receptor TfR1 in two brain regions including the medial prefrontal cortex (mPFC) and the hippocampus (Figure 5A–F). Accordingly, we found that the expression of TF was significantly increased in both mPFC and hippocampus regions (*p* < 0.05, *n* = 4) (Figure 5B,E). In contrast, the upregulated expression of TfR1 was only detected in the mPFC region (Figure 5C), but not in the hippocampus (Figure 5F), suggesting the distinct expression and regulatory mechanisms for TfR1 in different brain regions. In addition, we also detected the local transcripts of TF in both mPFC and hippocampus. Interestingly, in agreement with the upregulated expression at the protein level, we found the expression of TF was also significantly upregulated in susceptible mice after CSDS compared to controls at transcriptional level, indicating the upregulated expression of TF in the brain might be contributed by both local transcription and circulated TF from peripheral tissues through BBB.

### 3.5. The expression of TF and TfR1 in Stressed Animals from the Gene Expression Omnibus (GEO) Database

Meanwhile, to further verify our results, we consulted and analyzed the RNA-sequencing data from GEO database and summarized the transcriptional levels of TF and TfR1 in multiple peripheral organs and different brain regions in various stress animal models, respectively (Table 3 and Table 4). Based on the RNA-seq database, the results further confirmed that the transcripts of TF and its receptor TfR1 gene were both significantly upregulated in multiple organs and various brain regions including amygdala, hippocampus and PFC etc (Table 3 and Table 4). Moreover, the upregulated expression of TF and TfR1 in peripheral organs, such as blood and spleen, and central brain can be found in various depression mouse models including CSDS, PTSD, handling and restraint-induced depression (Table 3 and Table 4). The upregulated transcription of TF and TfR1 in the brain indicates that in addition to the transported TF from the peripheral liver and blood into the brain, stress also significantly induced the upregulated transcription of TF and TfR1 in local brain areas.

Together, these results suggest that stress can significantly upregulate the expression of TF and its receptor TfR1 in the brain and peripheral organs such as liver and blood at both transcriptional and translational levels. The upregulated expression of TF in the brain after stress was probably contributed by local transcription and transportation from the peripheral tissues.

## 4. Discussion

Stresses are inevitable stimuli to all species to maintain their homeostasis. All mammals including humans and animals are severely hampered by various stresses [43]. Prolonged and severe stress can induce mental health disorders such as anxiety, depression, and post-traumatic stress disorders [44]. Multiple biomarkers from different tissues have been identified including proteins, hormones, metabolites, miRNA, etc., as potent indicators of various biological processes and consequences induced by different types of stress and stressors [45,46,47,48]. Different peripheral biomarkers may be responsible for identical type of stress, either generally or specifically. Therefore, it is essential to identify the promising and reliable biomarkers highly responsible for specific pathophysiological aspects of the particular stress, which might be critical for the early diagnosis and intervention of stress-induced mental disorders, such as depression [48,49,50,51,52].

In this study, by using 2-DE combined with mass spectrometry proteomic analysis technique, we compared the differentially expressed proteins from the liver lysates between control and susceptible mice after CSDS stress. We identified that the expression of transferrin (TF) was notably up-regulated in liver, blood, and multiple brain regions. Moreover, the expression of its receptor TfR1 was also significantly upregulated in susceptible mice in both liver and multiple brain regions. Given that TF is synthesized primarily in the liver and can cross the blood-brain barrier, our study provides a possible working model that TF and its receptor TfR1 might mediate the pathophysiological responses in stress-induced mental disorders, such as anxiety and depression.

Stress affects the homeostasis of the liver and central nervous system, probably by targeting and regulating the liver-brain axis [53]. Different types of stressors can induce the injury to liver, such as hypoxia-reoxygenation, over-activation of Kupffer cells, oxidative stress, influx of gut-derived lipopolysaccharide and norepinephrine, and over-production of stress hormones and activation of the sympathetic nerve [54]. Stress causes the release of a large amount of glucocorticoids and catecholamines, and these hormones will act on the catabolism process of the liver to induce the occurrence of inflammatory liver disease [55]. In this study, we identified and confirmed several differentially expressed proteins in CSDS livers compared to non-stressed controls, such as the upregulated expression of transferrin (TF), Cps1, and Hmgb1, and the downregulated expression of PMM2 (Figure 2B,C). Previous study shows the constitutive release of Cps1 in bile serving as a protective cytokine during acute liver injury [56]. In agreement with previous studies, here, we found the significant upregulation of Cps1 in CSDS liver. Hmgb1 functions as a damage-associated molecular pattern (DAMP) contributing to the pathogenesis of various inflammatory diseases and is highly induced during liver injury [57,58,59]. Hmgb1 has been shown to be a promising therapeutic target for acute liver failure [60]. Human tissues tested showed the highest expression of PMM2 in the pancreas and liver [61], and patients with PMM2 deficiency (PMM2-CDG, congenital disorders of glycosylation) have a broad and variable spectrum of clinical presentations with liver dysfunction [62]. These data indicate that the CSDS can lead to disturbance of liver metabolism in susceptible mice, which might be contributing to the stress-induced pathophysiological responses. However, further study is warranted to discover the pathophysiological role of dysregulation of PMM2 in CSDS-induced depression.

Interestingly, we found that the expression of transferrin (TF) was notably up-regulated in liver, blood, and multiple brain regions in CSDS-induced susceptible mice. Moreover, the expression of its receptor TfR1 was also significantly upregulated in both liver and multiple brain regions in depressive mice. TF is synthesized primarily in the liver and can cross the blood-brain barrier (BBB), and depression and vulnerability to chronic social stress are associated with loss of this barrier integrity [63]. Thus, our study provides a possible working model that peripheral TF transportation and its receptor TfR1 in the central nervous system might mediate the pathophysiological responses in stress-induced mental disorders, such as anxiety and depression. Recently, iron metabolism and transport in the brain have been shown to be highly related to stress and stress-induced depression [64,65,66]. Stress-induced depression in the mouse model exhibits iron overload in the brain and excess iron intake might be a risk factor for cognitive impairment in depression patients [67].

For humans, there is also a link between iron deficiency and depression [68,69]. In 1995, studies proved that the levels of TfR1 in the plasma of patients with mental illness, especially schizophrenia and severe depression, were higher than those in the control [70]. The plasma TfR1 level in patients with bipolar disorder is also elevated and can be used as a biomarker for predicting mood disorder [71]. Moreover, both iron deficiency and iron overload can lead to the degeneration of dopaminergic neurons, thereby causing Parkinson’s disease [72]. Notably, the non-motor symptoms of Parkinson’s disease include depression, anxiety, sleep disorders and other mental diseases. The evaluation of iron metabolism and the degree of depression and anxiety in Parkinson’s patients demonstrated that serum iron content was negatively correlated with the degree of anxiety and depression, while serum concentration of TF was positively correlated with the degree of anxiety and depression [73], which is also consistent with our results.

In sum, in this study by using a CSDS-induced depressive mouse model in combination with proteomic analysis, we identified several peripheral stress-related biomarkers in liver including TF and its receptor TfR1. The content of TF was also dramatically increased in blood and central brain in CSDS stress-induced susceptible mice. Given that the expression of TfR1 was also significantly upregulated in the brain in stressed mice, our findings together with previous studies provide an interesting hypothesis that stress induced upregulation of TF in the liver result in TF transport to the brain through the blood-brain barrier, regulating the pathophysiological process of stress-induced depression.

## 5. Conclusions

We found the significantly upregulated expression of TF and its receptor TfR1 in both peripheral liver and the central brain in CSDS-induced depressive mouse model. Chronic social stress-induced BBB integrity deficiency might facilitate the transportation and accumulation of peripheral blood TF into the central brain regions including the amygdala, hippocampus, habenular, and prefrontal cortex. TF and TfR1 mediated iron metabolism and transport deficiency may contribute to the pathophysiological process of depression, and therefore may be a potential novel therapeutic target for stress-induced mental disorders.

## 6. Limitations and Future Directions

Although this study has provided several lines of evidence to support the potential role and mechanisms of TF and its receptor TfR1 in liver and brain during stress-induced depression, the detailed signaling and neurobiological mechanisms underlying TF-TfR1 interaction still need further investigation. For example, even though we demonstrated that TF in peripheral blood can cross the BBB into the brain under physiological conditions, whether this permeability is increased in CSDS-induced depressive mice still needs further characterization. Furthermore, the relationship and neurobiological mechanisms underlying upregulated TF-TfR1 signaling in the brain and impaired synaptic and brain circuitry are still unclear, and also need further in-depth exploration in the future.

## Figures and Tables

**Figure 1 brainsci-12-01267-f001:**
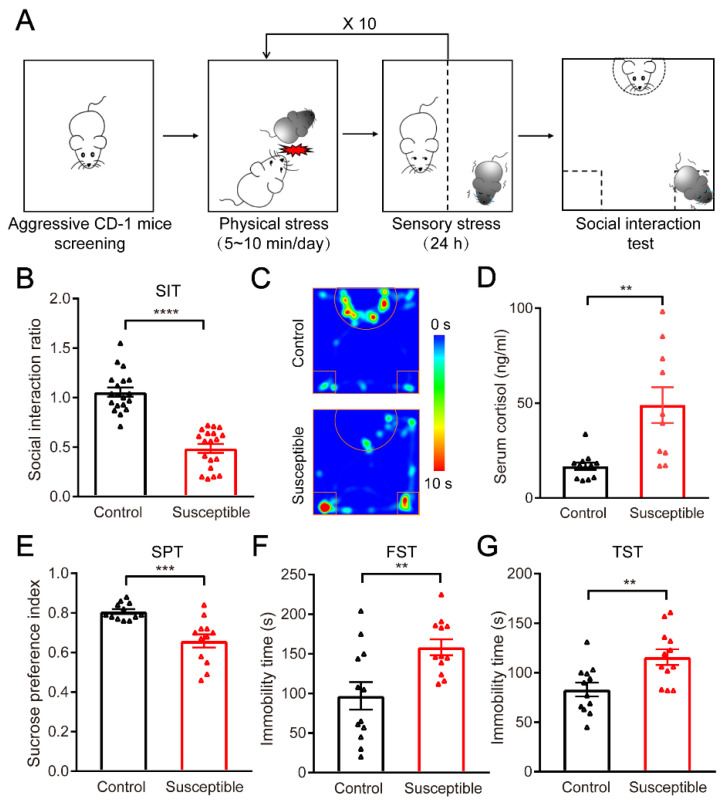
Chronic social defeat stress (CSDS) paradigm and depression-like phenotype in susceptible mice. (**A**) Experimental design of chronic social defeat stress (CSDS) using the resident intruder paradigm. (**B**) The social interaction ratio of susceptible mice compared to non-stressed controls (*n* = 19), *p* < 0.0001. (**C**) Representative trajectory heat maps of social interaction test between non-stressed controls (top) and susceptible mice (bottom). (**D**) The concentration of serum cortisol in non-stressed controls (*n* = 10) and susceptible mice (*n* = 12), *p* = 0.016. (**E**) The sucrose water preference test (SPT) in non-stressed controls (*n* = 12) and susceptible mice (*n* = 12), *p* = 0.0004. (**F**) The immobility time of the forced swimming test (FST) in non-stressed controls (*n* = 12) and susceptible mice (*n* = 12), *p* = 0.0059. (**G**) The immobility time of the tail-suspension test (TST) in non-stressed controls (*n* = 12) and susceptible mice (*n* = 12), *p* = 0.005. Unpaired *t* test. All data represent mean ± SEM. ** *p* < 0.01, *** *p* < 0.001, **** *p* < 0.0001.

**Figure 2 brainsci-12-01267-f002:**
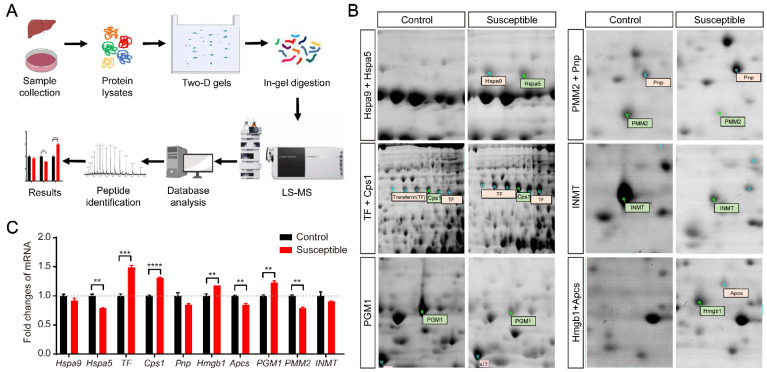
Identification of differential expression of protein in the livers in CSDS induced susceptible mice. (**A**) Schematic diagram of two-dimensional electrophoresis and mass spectrometry. (**B**) Representative 2-DE gel pattern of differentially expressed protein between non-stressed control and susceptible mice indicated by identical names, respectively. (**C**) qRT-PCR analysis of differentially expressed proteins at transcriptional level in liver RNA extractions from non-stressed controls (*n* = 3) and susceptible mice (*n* = 3). *Hspa9*: heat shock protein 9, *Hspa5*: heat shock protein 5, *TF*: transferrin, *Cps1*: carbamoyl phosphate synthase 1, *Pnp*: purine nucleoside phosphorylase, *Hmgb1*: High mobility group protein 1, *Apcs*: serum amyloid P component, *PGM1*: phosphoglucosidase-1, *PMM2*: phosphomannomutase 2, *INMT*: indoleethylamine N-methyltransferase. Unpaired *t* test. All data represent mean ± SEM. ** *p* < 0.01, *** *p* < 0.001, **** *p* < 0.0001.

**Figure 3 brainsci-12-01267-f003:**
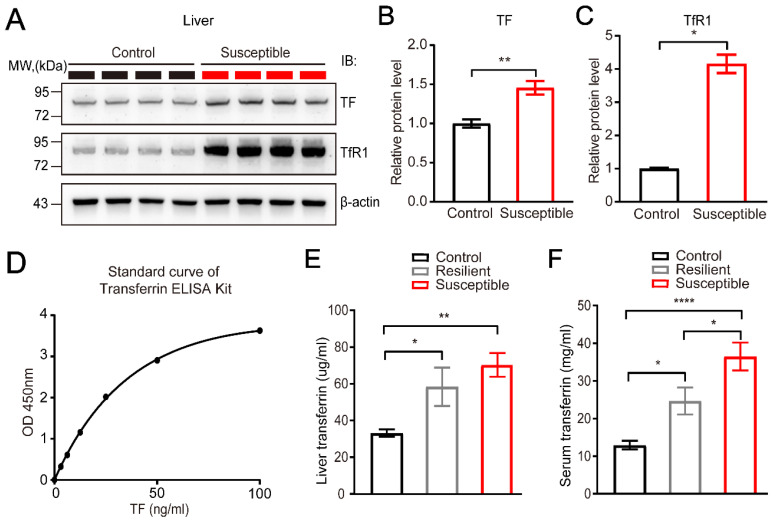
The elevated expression of TF and TfR1 in peripheral tissue induced by CSDS. (**A**–**C**) Representative immunoblot and quantitative analysis showing the elevated expression of TF and its receptor TfR1 in susceptible liver lysate induced by CSDS, normalized to β-actin. CSDS induced the significantly upregulated expression of TF (*p* = 0.0040) and TfR1 (*p* = 0.0286) in liver in susceptible mice (*n* = 4) compared to non-stressed controls (*n* = 4). (**D**) Representative standard curve of TF ELISA detection kit. (**E**) The concentration of TF in liver lysates detected by ELISA kit from non-stressed controls (*n* = 10), resilient mice (*n* = 10), and susceptible mice (*n* = 10), (F = 6.934, *p* = 0.0037), multiple comparison (C-S, *p* = 0.0031). (**F**) The concentration of TF in the serum detected by ELISA kit from non-stressed controls (*n* = 10), resilient mice (*n* = 10), and susceptible mice (*n* = 10), (F = 14.9, *p* < 0.0001), multiple comparison (Control vs. Resilient, *p* = 0.0296; Control vs. Susceptible, *p* < 0.0001; Resilient vs. Susceptible, *p* = 0.0280). All data represent mean ± SEM. * *p* < 0.05, ** *p* < 0.01, **** *p* < 0.0001.

**Figure 4 brainsci-12-01267-f004:**
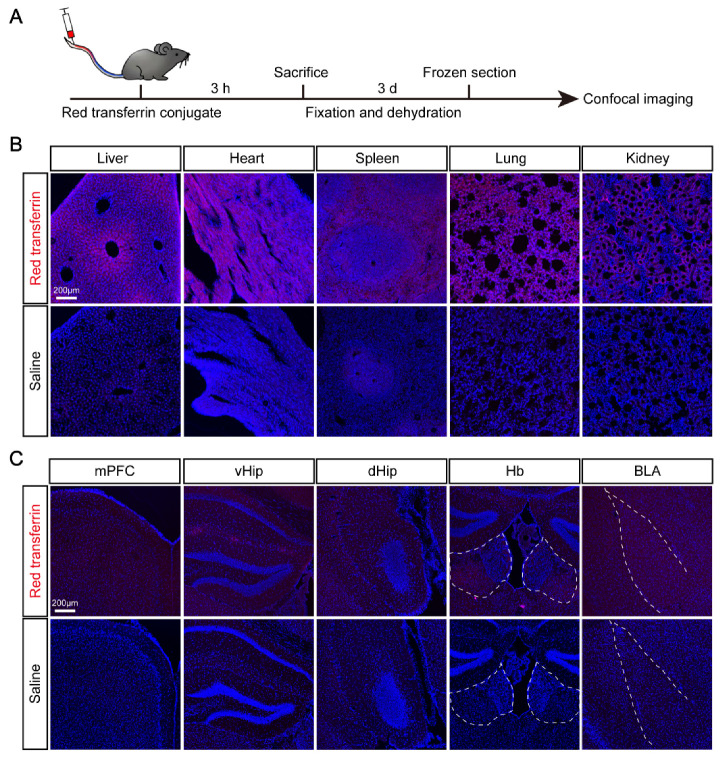
Delivery of exogenous TF peripherally across the blood brain barrier into the brain. (**A**) Schematic diagram of red transferrin conjugate delivery and imaging. (**B**) Red fluorescence signals were detected in multiple tissues including liver, heart, spleen, lung and kidney of injected mice (*n* = 3). (**C**) Red fluorescence signals were detected in the central brain including mPFC, vHip, dHip, Hb and BLA of injected mice (*n* = 3). mPFC, medial prefrontal cortex; vHip, ventral hippocampus; dHip, dorsal hippocampus; Hb: habenular; BLA, basolateral amygdala.

**Figure 5 brainsci-12-01267-f005:**
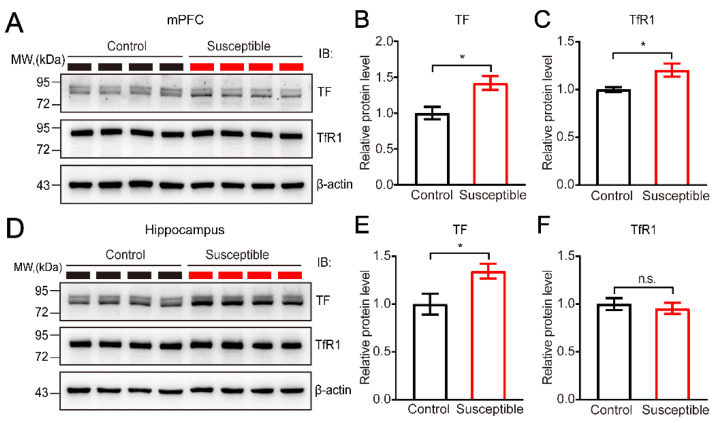
Elevated expression of TF in the mPFC and hippocampus induced by CSDS. (**A**–**C**) Representative immunoblot and quantitative analysis showing the elevated expression of TF and its receptor TfR1 in mPFC induced by CSDS, normalized to β-actin. CSDS induced the significantly upregulated expression of TF (*p* = 0.0179) and TfR1 (*p* = 0.0308) in mPFC in susceptible mice (*n* = 4) compared to non-stressed controls (*n* = 4). (**D**–**F**) Representative immunoblot and quantitative analysis showing the elevated expression of TF, but not its receptor TfR1 in the hippocampus induced by CSDS, normalized to β-actin. CSDS induced the significantly upregulated expression of TF (*p* = 0.0415) in the hippocampus in susceptible mice (*n* = 4) compared to non-stressed controls (*n* = 4). The expression of TfR1 did not change significantly (*p* = 0.6172) in the hippocampus in susceptible mice (*n* = 4) compared to non-stressed controls (*n* = 4). Unpaired *t* test. All data represent mean ± SEM. * *p* < 0.05, n.s., not significant.

**Table 1 brainsci-12-01267-t001:** Primers used for qRT-PCR.

Gene	Sequences (5′ to 3′)	Product Size
*Hspa9*	F: ATGGCTGGAATGGCCTTAGC	101 bp
	R: ACCCAAATCAATACCAACCACTG	
*Hspa5*	F: ACTTGGGGACCACCTATTCCT	134 bp
	R: ATCGCCAATCAGACGCTCC	
*Tf*	F: TGGGGGTTGGGTGTACGAT	103 bp
	R: AGCGTAGTAGTAGGTCTGTGG	
*Cps1*	F: ACATGGTGACCAAGATTCCTCG	119 bp
	R: TTCCTCAAAGGTGCGACCAAT	
*Pnp*	F: ATCTGTGGTTCCGGCTTAGGA	194 bp
	R: TGGGGAAAGTTGGGTATCTCAT	
*Hmgb1*	F: GGCGAGCATCCTGGCTTATC	86 bp
	R: GGCTGCTTGTCATCTGCTG	
*Apcs*	F: AGACAGACCTCAAGAGGAAAGT	117 bp
	R: AGGTTCGGAAACACAGTGTAAAA	
*PGM1*	F: CAGAACCCTTTAACCTCTGAGTC	137 bp
	R: CGAGAAATCCCTGCTCCCATAG	
*PMM2*	F: TGGTAGGTGGGTCAGATTTTGA	93 bp
	R: CCAAGCCATTCTCTGGAAACA	
*INMT*	F: GCAGAGCAGGAAATCGTAAAGT	167 bp
	R: GGGGTGTAGTCAGTGACAATGAT	
*Actb1*	F: GTCCCAGACATCAGGGAGTAA	102 bp
	R: TCGGATACTTCAGCGTCAGGA	

*Hspa9*: heat shock protein 9, *Hspa5*: heat shock protein 5, *TF*: transferrin, *Cps1*: carbamoyl phosphate synthase 1, *Pnp*: purine nucleoside phosphorylase, *Hmgb1*: high mobility group protein 1, *Apcs*: serum amyloid P component, *PGM1*: phosphoglucosidase-1, *PMM2*: phosphomannomutase 2, *INMT*: indoleethylamine N-methyltransferase.

**Table 2 brainsci-12-01267-t002:** Summary of the identity and related information of the differentially expressed proteins between control and CSDS livers.

Name	UniProt Code	ProtScore	Mass	Matches	Sequences
Hspa9	P38647	1817	73,701	45	19
Hspa5	P20029	711	72,492	24	14
TF	Q921l1	7030	78,841	204	34
Cps1	Q8C196	2995	165,711	85	43
PGM1	Q9D0F9	1366	61,665	34	19
PMM2	Q9Z2M7	1265	27,981	38	10
Pnp	P23492	8749	32,541	223	16
INMT	P40936	1401	30,068	41	9
Hmgb1	P63158	933	25,049	20	7
Apcs	P12246	1122	26,401	30	5

**Table 3 brainsci-12-01267-t003:** The expression of TF in different brain regions and tissues in various stressed animal models from GEO database.

GEO Accession	Models	Tissues	logFC	*p* Value
GSE151807	CMS	amygdala	−0.273	0.0010
GSE151807	CMS	hippocampus	0.296	0.0013
GSE151807	CMS	PFC	−0.099	0.0035
GSE151807	CMS	cerebral cortex	0.419	0.0016
GSE100236	Handling	dorsal hippocampus	0.060	0.4546
GSE100236	Handling	ventral hippocampus	0.144	0.0305
GSE100236	Restraint	dorsal hippocampus	0.177	0.0555
GSE100236	Restraint	ventral hippocampus	0.071	0.1506
GSE100236	FS	dorsal hippocampus	0.246	0.0006
GSE100236	FS	ventral hippocampus	0.033	0.5636
GSE172451	CSI	ventral hippocampus	0.051	0.0418
GSE172451	CSI + RES	ventral hippocampus	0.103	0.0042
GSE180055	ELS	mPFC	0.039	0.1343
GSE68077	PTSD	blood	0.781	0.6360
GSE68077	CSDS	blood	2.812	0.0025
GSE68077	PTSD	spleen	2.095	0.0044
GSE68077	CSDS	spleen	1.339	0.1120
GSE68077	PTSD	heart	0.598	0.2926
GSE68077	CSDS	heart	−0.417	0.2320

CMS, chronic mild stress; FS, forced swimming; CSI, chronic social instability; RES, restraint stress; ELS, early life stress; PTSD, post-traumatic stress disorder; mPFC, medial prefrontal cortex; CSDS, chronic social defeat stress.

**Table 4 brainsci-12-01267-t004:** The expression of TfR1 in different brain regions and tissues in various stressed animal models from GEO database.

GEO Accession	Models	Tissue	logFC	*p* Value
GSE151807	CMS	amygdala	−0.523	0.0001
GSE151807	CMS	hippocampus	0.660	<0.0001
GSE151807	CMS	PFC	0.326	<0.0001
GSE151807	CMS	cerebral cortex	0.845	<0.0001
GSE100236	Handling	dorsal hippocampus	−0.144	0.13243
GSE100236	Handling	ventral hippocampus	−0.243	8.04 × 10^−5^
GSE100236	Restraint	dorsal hippocampus	−0.146	0.035729
GSE100236	Restraint	ventral hippocampus	−0.246	0.000285
GSE100236	FS	dorsal hippocampus	0.070	0.444395
GSE100236	FS	ventral hippocampus	−0.241	0.000132
GSE172451	CSI	ventral hippocampus	0.103	0.0115
GSE172451	CSI + RES	ventral hippocampus	0.131	0.0062
GSE180055	ELS	mPFC	0.062	0.0292
GSE68077	PTSD	blood	0.179	0.667
GSE68077	CSDS	blood	0.787	0.063435
GSE68077	PTSD	spleen	−0.388	0.225383
GSE68077	CSDS	spleen	0.392	0.615
GSE68077	PTSD	heart	−1.008	0.098956
GSE68077	CSDS	heart	−1.038	0.0052

CMS, chronic mild stress; FS, forced swimming; CSI, chronic social instability; RES, restraint stress; ELS, early life stress; PTSD, post-traumatic stress disorder; mPFC, medial prefrontal cortex; CSDS, chronic social defeat stress.

## Data Availability

Not applicable.

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
