# Peer review of "Identification and Characterization of Elevated Expression of Transferrin and Its Receptor TfR1 in Mouse Models of Depression"

_brainsci, 2022, doi:10.3390/brainsci12101267_

Round 1

Reviewer 1 Report

Comments and Suggestions for Authors

In this study, the authors identified and characterized the expression of TF and TfR1 in different organs and brain regions. They showed that TF and TfR1 are up-regulated and that TF can cross the BBB in chronic social defeat stress (CSDS)-induced depressed mice, which may contribute to the pathophysiology of depression. The results of this work are interesting, but some points need to be addressed for publication:

1.       2-DE: please clarify whether the strips were reduced and alkylated or they were run under non-reducing conditions.

2.       About reverse transcription: the authors do not mention the use of DNAse to avoid gDNA contamination or how they controlled this issue: i.e. whether it was through primer design or through DNAse treatment, please describe this in the methodology.

3.       Figure 2D (q-RT-PCR): please describe the method used to calculate the fold change; some methods are described in the work of Michael Pfaffl 2001 (PMID: 11328886; PMCID: PMC55695; DOI: 10.1093/nar/29.9.e45).

4.       Figure 2B: the identification numbers of the 2DE GEL are poorly defined; it would be much better if this information could be improved with clearer numbers.

5.       Figure 2C: What type of score is it? mascot score? I suggest presenting this information as Table and not as image and include UNIPROT code.

6.       Figure 2c presents more proteins that were detected to be altered, but they are little discussed. The authors should discuss a little more on them even if they have not been subsequently validated.

7.       The authors demonstrated that TF in peripheral blood can cross the BBB into the brain under physiological conditions, which in turn may contribute to the pathophysiological process of stress-induced depression. It would have been interesting to know whether this permeability is increased in an animal under stress. However, the authors could discuss in more depth what is known about BBB permeability under stress conditions, which could extend the information in this regard.

8. methods: please indicate how many animals were used in each test.

Author Response

Reviewer #1:

In this study, the authors identified and characterized the expression of TF and TfR1 in different organs and brain regions. They showed that TF and TfR1 are up-regulated and that TF can cross the BBB in chronic social defeat stress (CSDS)-induced depressed mice, which may contribute to the pathophysiology of depression. The results of this work are interesting, but some points need to be addressed for publication:

  1. 2-DE: please clarify whether the strips were reduced and alkylated or they were run under non-reducing conditions.

Thank you for your great question. The 2-DE-gel proteins were reduced in order to break disulphide bonds and alkylated to prevent the bonds reforming as previously described (Herbert et al. Electrophoresis, 2001, 22: 2046-2057).

  1. About reverse transcription: the authors do not mention the use of DNAse to avoid gDNA contamination or how they controlled this issue: i.e. whether it was through primer design or through DNAse treatment, please describe this in the methodology.

Thank you for your comment. In this study, to reduce or eliminate gDNA contamination in RNA preparations, DNase I treatment was used for removing DNA contamination from RNA samples to ensure that gDNA is minimized before real-time RT-PCR analysis.

  1. Figure 2D (q-RT-PCR): please describe the method used to calculate the fold change; some methods are described in the work of Michael Pfaffl 2001 (PMID: 11328886; PMCID: PMC55695; DOI: 10.1093/nar/29.9.e45).

The detailed method used to calculate the fold changes of the qRT-PCR and the references including the reviewer suggested were added in the revised manuscript. Thank you!

  1. Figure 2B: the identification numbers of the 2DE GEL are poorly defined; it would be much better if this information could be improved with clearer numbers.

We apologize for the inaccurate description. The identity of the 2-DE-gel samples were renamed in the revised Fig. 2B.

  1. Figure 2C: What type of score is it? mascot score? I suggest presenting this information as Table and not as image and include UNIPROT code.

Thank you for the comment and suggestion. The score listed in Fig 2C is ProtScore which has been added in the revised Table 2. This information including UNIPROT codes has been presented as a separated Table 2 in the revised manuscript.

  1. Figure 2c presents more proteins that were detected to be altered, but they are little discussed. The authors should discuss a little more on them even if they have not been subsequently validated.

The differentially expressed proteins identified but not subsequently validated in this study were discussed in the revised manuscript. Thank you for the great suggestion!

  1. The authors demonstrated that TF in peripheral blood can cross the BBB into the brain under physiological conditions, which in turn may contribute to the pathophysiological process of stress-induced depression. It would have been interesting to know whether this permeability is increased in an animal under stress. However, the authors could discuss in more depth what is known about BBB permeability under stress conditions, which could extend the information in this regard.

Thank you for the great suggestion! We totally agree with the reviewer’s comment that the altered or increased permeability of BBB and TF under stress conditions might be informative to understand the pathophysiological process of stress-induced depression. Therefore, we added more related references and discussions to reinforce this notion in the revised manuscript.

  1. methods: please indicate how many animals were used in each test.

The animal numbers used in each test have been described in the figure legends previously. To address the reviewer’s comment, we have added the information in each test within the “Materials and Methods” part in the revised manuscript. Thank you!

Reviewer 2 Report

Comments and Suggestions for Authors

24 August 2022 

Manuscript ID: brainsci-1885576

Type: Article

Title: ‘dentification and Characterization of Elevated Expression of Transferrin and its Receptor TfR1 in Mouse Models of Depression’ by Chang X et al., submitted to Brain Sciences 

Dear Authors, 

The present study entitled ‘Identification and Characterization of Elevated Expression of Transferrin and its Receptor TfR1 in Mouse Models of Depression’ is a well-written and useful summary of the status of knowledge on the potential roles of transferrin protein (TF) in the liver and its receptor TfR1 in the brain in stress-induced depression. Here authors, using a social defeat stress (CSDS) mouse model, found a significant upregulation of TF in the liver, the peripheral blood, and the brain in CSDS-exposed mice. Furthermore, bioinformatics analysis of the GEO database from various mouse models of depression revealed a significant upregulated transcripts of TF and its receptor TfR1 in multiple brain regions, including amygdala, hippocampus and prefrontal cortex. 

The main strength of this manuscript is that it addresses an interesting and timely question, describing the potential role and mechanisms of liver and brain in stress-induced depression. In general, I think the idea of this study is really interesting and the authors’ fascinating observations on this timely topic may be of interest to the readers of Brain Sciences. However, some comments, as well as some crucial evidence that should be included to support the author’s argumentation, needed to be addressed to improve the quality of the manuscript, its adequacy, and its readability prior to the publication in the present form, in particular reshaping parts of the Introduction and Methods sections by adding more evidence and theoretical constructs. 

Please consider the following comments:

1.         Abstract: Please proportionally present the abstract with 200 words including the background, the methods, the results, and the conclusion. Also, please expand the abbreviation ‘GEO’ and avoid showing the abbreviations in the parenthesis if they do not appear later in the abstract.  

2.         Keywords: Please consider adding ‘chronic social defeat stress (CSDS)’ as keyword. Also, please list ten keywords and use them as many as possible in the first two sentence of the abstract.

3.         A Graphical Abstract is highly recommended.

4.         In general, I recommend authors to use more references to back their claims, especially in the Introduction of the manuscript, which I believe is lacking. Thus, I recommend the authors to attempt to expand the topic of their article, as the bibliography is too concise: nonetheless, in my opinion, less than 50/60 articles for a research article are insufficient. Indeed, currently authors cite 44 papers, and they are too low. Therefore, I suggest the authors to focus their efforts on researching more relevant literature: I believe that adding more studies and reviews will help them to provide better and more accurate background to this study.

5.         The objectives of this study are generally clear and to the point; however, I believe that there are some ambiguous points that require clarification or refining. For example, I believe that here the authors should be explicit about the necessity to compare the differentially expressed proteins from the liver lysates between control and CSDS mice, and how these proteins are involved in stress-induced behavioral responses.

6.         Introduction: The ‘Introduction’ section is well-written and nicely presented, with a good balance of descriptive text and information about etiology and symptomatology of depressive disorder. Nevertheless, I believe that more information about pathophysiology and core features of this disorder will provide a better and more accurate background, because as it stands, this information is not highlighted in the text. In this regard, I would suggest adding more information on pathological neural substrates of depression, specifically on structural as well as functional abnormalities of specific brain regions (i.e., amygdala, hippocampus, dorsomedial thalamus, and prefrontal cortex), on related and on related effects on patients’ cognitive impairments, and the translational applicability to depression (https://doi.org/10.3390/biomedicines9101293). In my opinion, authors could further explore significant structural brain alterations and impaired brain circuits in depressive disorder (https://doi.org/10.1016/j.tins.2022.04.003; https://doi.org/10.1111/cns.12835; https://doi.org/10.1111/psyp.14122) and focus on relationship between the molecular regulation of higher-order neural circuits and neuropathological alterations in this psychiatric disorder (https://doi.org/10.3390/cells11162607; https://doi.org/10.3390/biomedicines9050517).

7.         Introduction, Lines 45-46: I advise the authors to present some citations.

8.         CSDS mice model: Could the authors provide the specific number of mice that were used in the experiments? Also, could they be explicit about the rationale behind the choice to compare outbred and inbred mouse strains?

9.         Behavioral experiments: This paragraph that explains how CSDS stressed mice have depression-like phenotype and behaviors is the most important part of the study and should clearly describe all the experimental sessions in detail; therefore, this section might be improved by including further explanations, allowing the effective communication of experimental procedures.

10.      Acquirement and analysis of RNA sequencing data: I suggest rewriting this section more accurately. Please provide more information about experimental details while performing RNA-Seq.

11.      In my opinion, I think the ‘Conclusions’ paragraph would benefit from some thoughtful as well as in-depth considerations by the authors, because as it stands, it is very descriptive but not enough theoretical as a discussion should be. Authors should make an effort, trying to explain the theoretical implication as well as the translational application of their research.

12.      In according to the previous comment, I would ask the authors to include a ‘Limitations and future directions’ section before the end of the manuscript, in which authors can describe in detail and report all the technical issues brought to the surface, also to state the potential of this study, the goal, the knowledge and technology required to achieve this goal. Consider, among the limitations, that proactive inhibitory control has not been assessed.

13.      Figures: Please change the scale of the vertical axis and use the same minimum/maximum scale value in all the graphs.

14.      References: Authors should consider revising the bibliography, as there are several incorrect citations. Indeed, according to the Journal’s guidelines, they should provide the volume number in italics for all the references.

Overall, the manuscript contains three tables, five figures and 44 references. The number of references is too low for an original research article, and this prevents the possibility of publishing it in this form – in my opinion. However, the manuscript might carry important value describing the potential role and mechanisms of liver and brain in stress-induced depression.

I hope that, after these careful revisions, this paper can meet the Journal’s high standards for publication. I am available for a new round of revision of this paper. I declare no conflict of interest regarding this manuscript.

Best regards,

Reviewer

Author Response

Reviewer #2:

The present study entitled ‘Identification and Characterization of Elevated Expression of Transferrin and its Receptor TfR1 in Mouse Models of Depression’ is a well-written and useful summary of the status of knowledge on the potential roles of transferrin protein (TF) in the liver and its receptor TfR1 in the brain in stress-induced depression. Here authors, using a social defeat stress (CSDS) mouse model, found a significant upregulation of TF in the liver, the peripheral blood, and the brain in CSDS-exposed mice. Furthermore, bioinformatics analysis of the GEO database from various mouse models of depression revealed a significant upregulated transcripts of TF and its receptor TfR1 in multiple brain regions, including amygdala, hippocampus and prefrontal cortex. 

The main strength of this manuscript is that it addresses an interesting and timely question, describing the potential role and mechanisms of liver and brain in stress-induced depression. In general, I think the idea of this study is really interesting and the authors’ fascinating observations on this timely topic may be of interest to the readers of Brain Sciences. However, some comments, as well as some crucial evidence that should be included to support the author’s argumentation, needed to be addressed to improve the quality of the manuscript, its adequacy, and its readability prior to the publication in the present form, in particular reshaping parts of the Introduction and Methods sections by adding more evidence and theoretical constructs. 

Please consider the following comments:

  1. Abstract: Please proportionally present the abstract with 200 words including the background, the methods, the results, and the conclusion. Also, please expand the abbreviation ‘GEO’ and avoid showing the abbreviations in the parenthesis if they do not appear later in the abstract.

Thank you for the suggestion. We have carefully revised the abstract according to your suggestions.

  1. Keywords: Please consider adding ‘chronic social defeat stress (CSDS)’ as keyword. Also, please list ten keywords and use them as many as possible in the first two sentence of the abstract.

    In fact, chronic social defeat stress (CSDS) has been listed as the keyword already. In the revised manuscript, we have added additional more keywords. Thank you!

  1. A Graphical Abstract is highly recommended.

    This paper does not need a graphical abstract. Thank you!

  1. In general, I recommend authors to use more references to back their claims, especially in the Introduction of the manuscript, which I believe is lacking. Thus, I recommend the authors to attempt to expand the topic of their article, as the bibliography is too concise: nonetheless, in my opinion, less than 50/60 articles for a research article are insufficient. Indeed, currently authors cite 44 papers, and they are too low. Therefore, I suggest the authors to focus their efforts on researching more relevant literature: I believe that adding more studies and reviews will help them to provide better and more accurate background to this study.

    We have expanded and added more references in the revised Introduction part to amplify the abundancy of the background and bibliography of the manuscript. Thank you!

  1. The objectives of this study are generally clear and to the point; however, I believe that there are some ambiguous points that require clarification or refining. For example, I believe that here the authors should be explicit about the necessity to compare the differentially expressed proteins from the liver lysates between control and CSDS mice, and how these proteins are involved in stress-induced behavioral responses.

    Thank you for your comments and suggestions. We have explained the necessity to compare the differentially expressed proteins from the liver lysates between control and CSDS mice in the revised manuscript.

  1. Introduction: The ‘Introduction’ section is well-written and nicely presented, with a good balance of descriptive text and information about etiology and symptomatology of depressive disorder. Nevertheless, I believe that more information about pathophysiology and core features of this disorder will provide a better and more accurate background, because as it stands, this information is not highlighted in the text. In this regard, I would suggest adding more information on pathological neural substrates of depression, specifically on structural as well as functional abnormalities of specific brain regions (i.e., amygdala, hippocampus, dorsomedial thalamus, and prefrontal cortex), on related and on related effects on patients’ cognitive impairments, and the translational applicability to depression (https://doi.org/10.3390/biomedicines9101293). In my opinion, authors could further explore significant structural brain alterations and impaired brain circuits in depressive disorder (https://doi.org/10.1016/j.tins.2022.04.003; https://doi.org/10.1111/cns.12835; https://doi.org/10.1111/psyp.14122) and focus on relationship between the molecular regulation of higher-order neural circuits and neuropathological alterations in this psychiatric disorder (https://doi.org/10.3390/cells11162607; https://doi.org/10.3390/biomedicines9050517).

Thank you for the comments and suggestions. The pathological features including structural and functional abnormalities of specific brain regions and brain circuits in depressive disorders and related references were added in the revised Introduction part of the manuscript.

  1. Introduction, Lines 45-46: I advise the authors to present some citations.

The references were provided in the revised manuscript. Thank you!

  1. CSDS mice model: Could the authors provide the specific number of mice that were used in the experiments? Also, could they be explicit about the rationale behind the choice to compare outbred and inbred mouse strains?

Thank you for the comments and suggestions. The specific number of mice that were used in the experiments were provided in the revised manuscript. The rationale behind the choice to use inbred C57BL/6J mouse strain was also explained accordingly.

  1. Behavioral experiments: This paragraph that explains how CSDS stressed mice have depression-like phenotype and behaviors is the most important part of the study and should clearly describe all the experimental sessions in detail; therefore, this section might be improved by including further explanations, allowing the effective communication of experimental procedures.

We thank the reviewer for the comment. More detailed description and explanations about the CSDS stressed mice performance in behavioral assays were added in the revised manuscript. Thank you!

  1. Acquirement and analysis of RNA sequencing data: I suggest rewriting this section more accurately. Please provide more information about experimental details while performing RNA-Seq.

It should be noted that we did not perform the RNA-seq experiment. All the RNA-seq data used in this study were collected from the previously reported database. Nevertheless, we have strengthened this point in the revised manuscript to increase the clarity and accuracy of the information.

  1. In my opinion, I think the ‘Conclusions’ paragraph would benefit from some thoughtful as well as in-depth considerations by the authors, because as it stands, it is very descriptive but not enough theoretical as a discussion should be. Authors should make an effort, trying to explain the theoretical implication as well as the translational application of their research.

    Thank you for the suggestion. We have revised the Conclusions section to make it sounds in-depth and be more theoretical.

  1. In according to the previous comment, I would ask the authors to include a ‘Limitations and future directions’ section before the end of the manuscript, in which authors can describe in detail and report all the technical issues brought to the surface, also to state the potential of this study, the goal, the knowledge and technology required to achieve this goal. Consider, among the limitations, that proactive inhibitory control has not been assessed.

Added. Thank you.

  1. Figures: Please change the scale of the vertical axis and use the same minimum/maximum scale value in all the graphs.

Given that different graphs have different maximum value and number of the parameter in the vertical axis for each figure, therefore it is impossible for us to use the same scale value in all the graphs. Hope you can understand this point.

  1. References: Authors should consider revising the bibliography, as there are several incorrect citations. Indeed, according to the Journal’s guidelines, they should provide the volume number in italics for all the references.

    Changed. Thank you!

Overall, the manuscript contains three tables, five figures and 44 references. The number of references is too low for an original research article, and this prevents the possibility of publishing it in this form – in my opinion. However, the manuscript might carry important value describing the potential role and mechanisms of liver and brain in stress-induced depression.

I hope that, after these careful revisions, this paper can meet the Journal’s high standards for publication. I am available for a new round of revision of this paper. I declare no conflict of interest regarding this manuscript.

Round 2

Reviewer 2 Report

Comments and Suggestions for Authors

16 September 2022 

Manuscript ID: brainsci-1885576

Type: Article

Title: ‘Identification and Characterization of Elevated Expression of Transferrin and its Receptor TfR1 in Mouse Models of Depression’ by Chang X et al., submitted to Brain Sciences

Dear Authors, 

I am pleased to see that the authors did an excellent work clarifying most of the comments I have raised in the previous round of the review session. Currently, this paper is a well-written, timely piece of research investigating the potential role and mechanisms of the liver and the brain in stress-induced depression. That said, I just suggest some minor points below, I believe, for the betterment of this manuscript to finalize my review session.

1.      Materials and Methods: I recommend that the authors avoid using the abbreviation in the subtitle such as CSDS.

2.      Introduction: This section has improved substantially. Nevertheless, I would suggest adding more information on pathological neural substrates of depression and related effects on patients’ cognitive impairments, and the translational applicability to depression (https://doi.org/10.1016/j.tins.2022.04.003).

3.      Figures: I suggest presenting figures immediately after paragraphs in which a figure is referred to.

Overall, the manuscript contains five figures, four tables, and 71 references. This is a timely and needed work studying a significant upregulation of transferrin and its receptor in the liver, the peripheral blood, and the brain in chronic social defeat stress mouse model of mice. Thus, I believe that manuscript now meets the Journal’s standards for publication. I am always available for other reviews of such interesting and important articles. I look forward to seeing further study on this issue by these authors in the future.

Best regards,

Reviewer

Author Response

Reviewer #2:

I am pleased to see that the authors did an excellent work clarifying most of the comments I have raised in the previous round of the review session. Currently, this paper is a well-written, timely piece of research investigating the potential role and mechanisms of the liver and the brain in stress-induced depression. That said, I just suggest some minor points below, I believe, for the betterment of this manuscript to finalize my review session.

Thank you for your positive comments and great suggestions!

  1. Materials and Methods: I recommend that the authors avoid using the abbreviation in the subtitle such as CSDS.

The abbreviation in the subtitle has been changed. Thank you!

  1. Introduction: This section has improved substantially. Nevertheless, I would suggest adding more information on pathological neural substrates of depression and related effects on patients’ cognitive impairments, and the translational applicability to depression (https://doi.org/10.1016/j.tins.2022.04.003).

Added. Thank you!

  1. Figures: I suggest presenting figures immediately after paragraphs in which a figure is referred to.

The position of the figures presented in the paper was produced by the journal editors. Thank you!

Overall, the manuscript contains five figures, four tables, and 71 references. This is a timely and needed work studying a significant upregulation of transferrin and its receptor in the liver, the peripheral blood, and the brain in chronic social defeat stress mouse model of mice. Thus, I believe that manuscript now meets the Journal’s standards for publication. I am always available for other reviews of such interesting and important articles. I look forward to seeing further study on this issue by these authors in the future.

Thank you!
